# Is Short-Term Exposure to PM_2.5_ Relevant to Childhood Kawasaki Disease?

**DOI:** 10.3390/ijerph18030924

**Published:** 2021-01-21

**Authors:** Jongmin Oh, Ji Hyen Lee, Eunji Kim, Soontae Kim, Hae Soon Kim, Eunhee Ha

**Affiliations:** 1Department of Occupational and Environmental Medicine, Ewha Womans University College of Medicine, Seoul 07804, Korea; Jongminoh@ewha.ac.kr (J.O.); olodii36@ewhain.net (E.K.); 2Department of Pediatrics, Ewha Womans University College of Medicine, Seoul 07804, Korea; leejihyen@ewha.ac.kr; 3System Health & Engineering Major in Graduate School (BK21 Plus Program), Ewha Womans University, Seoul 07804, Korea; 4Department of Environmental and Safety Engineering, Ajou University, Suwon 16499, Korea; soontaekim@ajou.ac.kr

**Keywords:** PM_2.5_, children, Kawasaki disease, case-crossover design

## Abstract

*Background:* Kawasaki disease (KD) is an acute febrile vascular disease of unknown cause that affects the whole body. KD typically occurs in infants under the age of five and is found mainly in East Asian countries. Few studies have reported on the relationship between the pollutant PM_2.5_ and KD, and the evidence remains irrelevant or insufficient. *Objectives:* We investigated the relationship between short-term exposure to PM_2.5_ and KD hospitalizations using data from Ewha Womans University Mokdong Hospital, 2006 to 2016. *Methods:* We obtained data from the hospital EMR (electronic medical records) system. We evaluated the relationship between short-term exposure to PM_2.5_ and KD hospitalizations using a case-crossover design. We considered exposures to PM_2.5_ two weeks before the date of KD hospitalization. We analyzed the data using a conditional logistic regression adjusted for temperature and humidity. The effect size was calculated as a 10 μg/m^3^ increase in PM_2.5_ concentration. We performed a subgroup analysis by sex, season, age group, and region. In the two-pollutants model, we adjusted SO_2_, NO_2_, CO, and O_3_, but the effect size did not change. *Results:* A total of 771 KD cases were included in this study. We did not find any statistically significant relationship between PM_2.5_ and children’s KD hospitalization (two-day moving average: odds ratio (OR) = 1.01, 95% confidence intervals (CI) = 0.95, 1.06; seven-day moving average: OR = 0.98, CI = 0.91, 1.06; 14-day moving average: OR = 0.93, CI = 0.82, 1.05). A subgroup analysis and two pollutant analysis also found no significant results. *Conclusion:* We did not find a statistically significant relationship between PM_2.5_ and children’s KD hospitalizations. More research is needed to clarify the association between air pollution, including PM_2.5_, and KD.

## 1. Introduction

Kawasaki disease (KD), also known as Kawasaki syndrome, is an acute febrile illness of unknown cause that is primarily seen in children younger than five years of age. The disease was first described in Japan by Tomisaku Kawasaki in 1967 [1]. The first cases outside of Japan were reported in Hawaii in 1976 [2]. KD causes swelling in medium-sized arterial walls throughout the body, leading to bilateral bulbar conjunctival injection, cervical lymphadenopathy, changes in extremities, including erythema and induration, polymorphous rash, and changes in the lips and oral mucosa, including erythema, cracking, and strawberry tongue. The inflammation tends to affect the coronary arteries, which supply blood to the heart muscle [3,4]. Most children recover from the disease within a few weeks, but in 15–30% of untreated cases, KD patients develop heart problems, including coronary artery aneurysms, arrhythmias, and heart failure [5].

The overall rate of KD incidence has been increasing in recent years, but the incidence is highest among children who live in East Asia or who are of Asian ancestry. East Asian children have a 10–15-fold higher risk of developing KD than Caucasian children. The incidence of KD in Japan is the highest (264.8 per 100,000 children), followed by Korea (132.4 per 100,000 children) and Taiwan (82.77 per 100,000 children) [6].

Although several theories have been proposed regarding the etiology of KD, including environmental toxin exposure, autoimmune pathogenesis, and infectious diseases, it remains unknown. KD seems to be multifactorial and may occur in genetically susceptible individuals with an aberrant immune response to certain environmental triggers. The known genetic factors do not explain why disease incidence has increased in recent years. Thus, epidemiological studies have focused on potential environmental risk factors.

The incidence of KD has a striking seasonal variation, with different patterns in several countries. Previous studies have suggested that windborne environmental factors may trigger the disease in genetically susceptible children. The results of one study conducted in Taiwan did not support a positive association between ozone exposure and KD occurrence [7], and a U.S. study had similar findings [8]. However, in Japan, Takashi and colleagues reported that prenatal exposure to particulate air pollution may be associated with an increased risk of KD occurrence in children aged 6 to 30 months [9]. Whether fine particulate matter (PM_2.5_) contributes to KD remains unclear owing to limited monitoring data. This study analyzed the association between PM_2.5_ and the hospitalization of KD in children in South Korea over 10 years (2006–2016).

## 2. Materials and Methods

### 2.1. Kawasaki Disease Inpatient Definitions

We used the Electronic Medical Record (EMR) system hospitalization data of Ewha Womans University Mokdong Hospital. We obtained the raw data for a total of 3205 hospitalized KD cases from January of 2006 to December of 2016. The raw data included the patient’s sex, age, address, outpatient/inpatient records, prescriptions, and examination records. All KD patients have M30.3 codes (Mucocutaneous lymph node syndrome) based on the International Classification of Diseases code (ICD-10 code).

We used the following criteria to define KD patients: (1) duplicate data were deleted if the hospitalization date and discharge date were the same; (2) patients were excluded if they did not have an intravenous immunoglobulin (IVIG) injection prescribed; (3) the patient’s sex, age, and residence information was excluded if it was missing; (4) only children 10 years of age or younger were included; and (5) we limited the study area to three regions adjacent to the hospital (Seoul, Gyeonggi, and Incheon).

The study was reviewed and approved by the Ewha Womans University Seoul Hospital Institutional Review Board (IRB File No: SEUMC 2020-12-018-002).

### 2.2. Environmental Variable

PM_2.5_ measurements did not begin in South Korea until 2015. We used the estimated CMAQ (Community Multiscale Air Quality) PM_2.5_ data 2006–2016. CMAQ is the most widely used EPA-designed model in Korea. Due to a lack of information on PM_2.5_ concentrations before 2015, PM_2.5_ data were calculated by applying ground and phase observations and CMAQ modeling data. We used Weather Research Forecasting (WRF) version 3.6.1 (https://esrl.noaa.gov/gsd/wrfportal/WRFPortal.html) as input to the reanalyzed meteorological data to estimate exposure [10] and Sparse Matrix Operator Kernel Emissions version (SMOKE) version 3.1 (https://www.cmascenter.org/download/forms/step_1.cfm) as the emission-generation model. The modeling area of the chemical transport modeling system was set to a 27 km horizontal resolution (174 × 128 grids) in East Asia and a 9 km resolution (67 × 82 grids) in the Korean peninsula [11]. FNL (Final Operational Earth Analysis Data; NCEP, 2000), reanalysis data from the U.S. Oceanic and Atmospheric Administration (NOAA), was used as meteorological initial condition data. Our exposure data were based on observations and we interpolated the missing data. Therefore, our exposure data are a fusion of observational and simulated data. A description of the modeling process has been described in previous studies [11,12,13,14,15]. We estimated this data in the domestic administrative districts and calculated it as the average daily data. The CMAQ PM_2.5_ data were linked using administrative-district data based on the patients’ place of residence. A total of 37 administrative districts are linked (Seoul: 17, Gyeonggi: 14, Incheon: 6).

We also obtained air pollution monitoring data (https://www.airkorea.or.kr/web) for two-pollutant models. We collected data on five air pollutants (particulate matters (PM_10_), sulfur dioxide (SO_2_), nitrogen dioxide (NO_2_), carbon monoxide (CO), and ozone (O_3_). Monitoring stations are installed in multiple locations by region. We selected and linked the nearest monitoring station from the KD patient’s place of residence.

We used the daily average temperature and humidity data as covariates. These variables were obtained from the Korea Meteorological Administration (KMA). Since the temperature and humidity data are open by city, the daily mean temperature and humidity were linked to each patient’s residence cities (Seoul, Gyeonggi, Incheon).

### 2.3. Study Design

To investigate the acute relationship between short-term exposure to PM_2.5_ and KD hospitalization, we considered the bi-directional time-stratification case-crossover design. In this design, the date settings for the control periods are the same year, month, and day of the weeks for the KD hospitalization date, but the weeks differ. Therefore, three or four controls can be assigned to one KD patient. In this study design, because each case or control periods is the same patient, personal characteristics such as sex and age are not considered in the short-term fluctuations. Instead, we considered weather variables as potential confounding factors (e.g., temperature and humidity).

### 2.4. Statistical Analysis

We performed a conditional logistic regression to determine the relationship between PM_2.5_ exposure and KD hospitalization. This is an extended logistic regression method that accounts for several control periods. We calculated the odds ratio (OR) and confidence interval (CI) per 10-unit increases in PM_2.5_. To account for the effects of various exposure lag days, lag days were considered from the date of KD hospitalization (current day) up to two weeks ago. We considered both the single lag (lag 0 to lag 14) effects and the moving average (lag 0–1 to lag 0–14) effects. We suggest the moving average effect as the main analysis in this study.

We considered sex (boys/girls), season (warm season/cold season), age groups (<1, 1–4, 5–10), and region (Seoul/Gyeonggi/Incheon) in the sub-group analysis. The warm seasons include April-September, and the cold season includes January-March and October-December.

We calculated a Spearman’s correlation between air pollutants before considering the two-pollutants model. If the correlation coefficient between the two exposure variables is too high (more than 0.7), it was excluded from the analysis. In the two-pollutants model, PM_2.5_ is the main exposure and other air pollutants (SO_2_, NO_2_, CO, O_3_) were adjusted.

All the data preprocessing and statistical analysis were performed used R statistical software (Ver. 4.0.0 R development Core Team, Vienna, Austria). The significance level of alpha was 0.05.

## 3. Results

### Main Results

A total of 771 KD cases were included in our research (Figure 1). Table 1 shows the areas where our study patients, including Seoul, Gyeonggi, and Incheon. The majority of participants live in Seoul. Detailed administrative areas for the study’s residential areas are presented in Appendix A. Table 2 shows summary statistics of the exposure data during the period 2006–2016. We compared environmental exposure levels on case and control peropds using t-test. There was no difference in PM_2.5_ concentration between case periods and control periods. Appendix A shows the average exposure concentration for the study area (each district) during the study period. The single-lag exposure effects from the current day (lag 0) to the day before the two-week mark (lag 14) are shown in Appendix A. The effect sizes appear to decrease as the delay grows longer. Table 3 shows the model results for the association between PM_2.5_ exposure and children’s KD hospitalizations. Figure 2 shows the effects of exposure to PM_2.5_ on the current day (Lag 0), two-, seven-, and 14-day moving average (Lag 0–2, Lag 0–6, Lag 0–13) age at KD hospitalization. We did not find any association between short-term exposure to PM_2.5_ and children’s KD hospitalization. Subgroup analysis also found no association.

Appendix A provides the summary statistics of the daily air pollution data measured by a monitoring station. The correlation between PM_2.5_ and other air pollutants is shown in Appendix A. The correlation coefficients are as follows: PM_10_ [r: 0.92], SO_2_ [r: 0.51], NO_2_ [r: 0.52], CO [r: 0.56], and O_3_ [r: −0.07]. PM_10_ was excluded from the two-pollutant model due to its high correlation with PM_2.5_. We found no association between PM_2.5_ and children’s KD hospitalizations (Table 4) in the two-pollutant models.

## 4. Discussion

The present study was conducted in Seoul, Gyeonggi, and Incheon, South Korea, over 10 years. Our findings suggest that short-term exposure to PM_2.5_ was not significantly associated with an increased hospitalization of KD. We believe this is the first attempt to assess the impacts of PM_2.5_ on KD in South Korea.

KD is an acute early childhood vasculitis of unknown etiology. Common KD complications include coronary artery abnormalities (CAAs), which are currently the leading cause of acquired heart disease in children. Most cases occur between the ages of six months and eight years. However, the exact mechanisms underlying KD development remain unclear. Young children inhale relatively large amounts of PM_2.5_ because their lungs are less developed than adults. Previous reports have suggested that short-term exposure to PM_2.5_ may damage endothelial cells, impair lung and vascular function, stimulate the onset of a systemic inflammatory response, increase oxidative stress, and induce cardiac ischemia and repolarization abnormalities [16,17,18]. Our study found no association between PM_2.5_ and KD hospitalizations.

The incidence of KD has a striking seasonal variation with a differing pattern in several countries. Burns et al. studied seasonal patterns of KD across 25 countries from 1970 to 2012 [19]. Their reports suggested that an environmental trigger could play a role in determining the seasonality of KD cases worldwide. KD cases peak in the winter and decrease in the late summer and fall in the extratropical northern hemisphere, including Japan, South Korea, and the U.S. [20]. In China, there is a peak in the summer/spring [21,22], while the highest incidence of KD is observed between May and June in the tropics and the extratropical southern hemisphere [19].

These seasonal patterns may be driven by differences in infectious disease activity, the presence of environmental allergens, and the independent influence of ambient temperature. Recent studies also suggest that tropospheric winds from northern China carry KD’s etiologic agents into Japan and that a higher level of postnatal suspended PM (SPM) exposure indicates a higher level of desert dust in some regions [23,24]. The direct inhalation of air pollutants can alter innate and adaptive immune responses, potentially inducing cytokine-mediated endothelial cell injury in genetically susceptible children [25]. Moreover, postnatal air pollution exposure could increase the risk of respiratory infections, triggering abnormal immune responses and leading to KD onset.

Previous studies found no association between particulate matter and KD. In a recent study, Hwang et al. used a Taiwanese health insurance database to identify 695 KD hospitalizations in children under five years of age between the years 2000–2010 [7]. They found an association between KD hospitalizations and ozone pollution during the summer months, possibly because children spend more time outdoors in the summertime. However, no significant associations were found with other pollutants, including nitrogen dioxide and coarse PM. Zhijing et al. evaluated the relationship between ambient air pollution, temperature, and KD in Shanghai, China [26].

The three major air pollutants PM_10_, NO_2_, and SO_2_ had a positive but statistically insignificant association with KD hospitalizations. It was suggested that short-term exposure to high temperatures may significantly increase KD hospitalizations, but the evidence linking air pollution and KD was limited. Furthermore, Zhijing and colleagues inferred that high temperatures may promote the spread of infectious agents, such as fungi and bacteria, contributing to high exposure levels. They also suggested that children tend to spend more time outdoors when the temperature is high or moderate, increasing exposure to infectious agents.

In the U.S., Zeft et al. reported no significant association between particulate matter exposure and KD occurrence in seven metropolitan regions [8]. However, this study focused on short-term exposure (a few days to a week). Takashi et al. suggested that the effects of long-term exposure (more than one month) on KD occurrence should be evaluated [9]. Using nationwide, population-based, longitudinal survey data from Japan that began in 2010, this group found that early-life exposure to particulate air pollution, especially during pregnancy, was associated with an increased risk of hospital admissions for KD during early childhood.

In Italy, Elena et al. found no evident correlation between PM_10_ exposure and KD onset, suggesting that an environmental agent carried by the wind from a specific direction could trigger KD in genetically-susceptible individuals [27]. KD onset was more likely to occur during periods of substantially warmer night-time temperatures in years with a prevailing southwesterly mean flow. Further, they suggested that certain wind conditions are more favorable for disease onset, which may be associated with one or more airborne agents.

Especially, since the beginning of the pandemic of coronavirus disease-2019 (COVID-19), a surge of SARS-Cov-2 patients with COVID-19 syndrome overlapping with KD, called multisystem inflammatory syndrome in children (MIS-C), has been reported in 2020 [28,29,30]. KD and MIS-C share several common symptoms, such as skin rash, lymphadenopathy, strawberry tongue, coronary artery dilatation and an elevation inflammatory biomarkers such as C-reactive protein, procalcitonin, ferritin [31]. Furthermore, several unique symptoms of MIS-C and KD are prevalent in children, and children have fewer pro-inflammatory cytokines secretion and more active innate immune response than adults [31]. Therefore, understanding that the KD is rather not associated with PM_2.5_ may provide us with new information regarding the pathogenesis of MIS-C in COVID-19, as well as the conserve.

The current study has two key strengths. First, as KD is clinically defined by the onset of fever, a well-defined event date can be studied, unlike most rheumatologic diseases. Rheumatologic diseases usually have an indolent course and an extended prodromal period. In South Korea, nationwide KD surveys began in 1994 using patient data from all resident-training hospitals between 1991–1993, and they have since been conducted every three years. Therefore, data covering patients who received intravenous immunoglobulin following a KD diagnosis (ICD-10 [International Statistical Classification of Diseases and Related Health Problems, 10th ed], M30.3) can be obtained in South Korea [32]. This approach provides more data on KD incidence when compared to nationwide survey methods in other countries because mandatory subscription to the national health insurance programs of South Korea covers more than 99% of the population. Further, response rates to a nationwide survey from 2009–2011 and 2012–2014 were 87% and 94.8%, respectively. Our study was also conducted in a country with the second highest KD rates. KD incidence in Northeast Asian countries, including Japan, South Korea is 10–30 times greater than rates in the U.S. and Europe.

This study has several limitations. First, we assigned administrative district-level PM_2.5_ exposure data and did not measure or estimate individual-level exposure, including proximity to major roads or the activities of each participant’s living place. Inaccurate exposure assignment may not properly reflect the relationship to health effects. Therefore, further studies need accurate individual level exposure measurements. Second, we did not evaluate the relationship between prenatal exposure to PM_2.5_ and KD occurrence. As discussed above, PM_2.5_ exposure during pregnancy could affect fetal lung development and induce epigenetic changes which may influence immune programming and organ development. Third, we did not consider the effects of other meteorological variables (e.g., wind speed) as confounding variables. Because our results were not statistically significant, future study should account for all of the confounding factors that could influence KD. Finally, the further studies including genetic susceptibility between air pollution and KD would be worthwhile to evaluate the immune environment and its response in the body. Because KD occurs most often in children from northeast Asia. Even in the United States, KD occurs most often among children of Asian-American or Pacific Island ancestry [33].

## 5. Conclusions

In conclusion, we found no association between short-term PM_2.5_ exposure and KD occurrence in South Korea. Future research should compare KD cases between locations in different parts of the world with a larger sample size and incorporate statistical models that can determine associations between several environmental factors, ambient temperatures, and KD occurrence. Cooperative efforts are needed to elucidate KD’s underlying pathological mechanisms. Future research on the causal relationship of air pollution to KD is needed.

## Figures and Tables

**Figure 1 ijerph-18-00924-f001:**
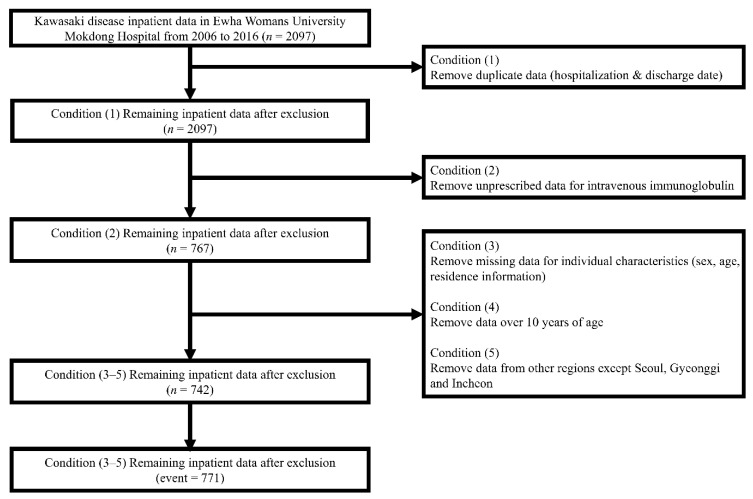
Defining Kawasaki cases at Ewha Womans University Mokdong Hospital during 2006–2016.

**Figure 2 ijerph-18-00924-f002:**
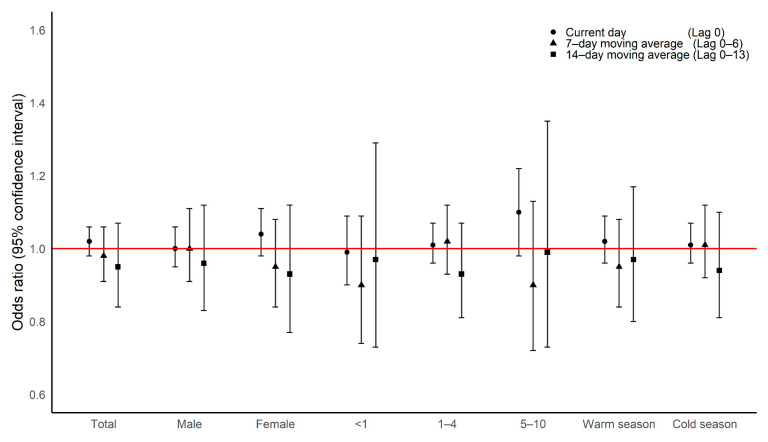
The association between current day, two-, seven- and 14-day moving average exposure to PM_2.5_ and children’s KD hospitalizations. The circle shape designates the effect size of the lag 0. The triangle shape indicates the effect size of the lag 0–1. The square shape indicates the effect size of the lag 0–6. The diamond shape shows the effect size of the lag 0–14. The red line demonstrates the borderline. The warm season includes the months of April to September, while the cold season includes October to March. All models were adjusted for daily mean temperature and humidity. The odds ratio was calculated per 10-unit increase.

**Table 1 ijerph-18-00924-t001:** Descriptive statistics on the characteristics of Kawasaki disease patients in Ewha Womans University Mokdong Hospital during the study period of 2006–2016.

Category	Non-Event	Event
Sex		
Boys	1549	458
Girls	1080	313
Season		
Warm season ^a^	1365	402
Cold season ^b^	1264	369
Age group		
<1	558	164
1–4	1729	505
5–10	342	102
Regions		
Seoul	2168	637
Gyeonggi	389	113
Incheon	72	21

^a^ The warm season includes April-September. ^b^ The cold season includes October-March.

**Table 2 ijerph-18-00924-t002:** Summary statistics for daily exposure variables during the study period of 2006–2016.

Exposure Variables	Case Periods (*n* = 771)		Control Periods (*n* = 2629)		Mean Difference	95% Confidence Interval
	Mean	SD	Mean	SD		
PM_2.5_ (μg/m^3^)	34.13	22.37	33.33	20.46	0.80	−0.96, 2.57
Mean temperature (°C)	12.82	10.93	12.70	11.19	0.12	−0.76, 1.00
Mean humidity (%)	62.57	14.83	62.34	14.86	0.23	−0.97, 1.41

Abbreviations: SD, standard deviation.

**Table 3 ijerph-18-00924-t003:** The association between short-term exposure to PM_2.5_ and children’s KD hospitalization, total and by sex, age group, season, and region.

		Sex		Age Group			Season		Regions		
Lag	Total	Boys	Girls	<1	1–4	5–10	Warm Season ^a^	Cold Season ^b^	Seoul	Gyeonggi	Incheon
0–1	1.01 (0.96, 1.06)	1.01 (0.95, 1.07)	1.02 (0.94, 1.10)	0.93 (0.82, 1.06)	1.02 (0.96, 1.08)	1.07 (0.95, 1.22)	1.00 (0.92, 1.09)	1.02 (0.95, 1.09)	1.00 (0.95, 1.06)	1.08 (0.94, 1.25)	0.97 (0.76, 1.24)
0–2	1.00 (0.95, 1.06)	1.01 (0.94, 1.08)	0.98 (0.89, 1.08)	0.92 (0.79, 1.06)	1.01 (0.95, 1.08)	1.03 (0.89, 1.19)	0.96 (0.88, 1.05)	1.03 (0.96, 1.11)	0.99 (0.93, 1.05)	1.11 (0.95, 1.29)	0.93 (0.68, 1.28)
0–3	0.99 (0.93, 1.05)	1.00 (0.93, 1.08)	0.96 (0.86, 1.06)	0.90 (0.76, 1.06)	1.00 (0.93, 1.08)	0.99 (0.83, 1.17)	0.94 (0.85, 1.04)	1.02 (0.95, 1.11)	0.97 (0.90, 1.03)	1.13 (0.95, 1.34)	0.94 (0.66, 1.34)
0–4	0.98 (0.92, 1.05)	1.00 (0.92, 1.09)	0.95 (0.85, 1.06)	0.91 (0.77, 1.07)	1.01 (0.93, 1.09)	0.95 (0.78, 1.15)	0.94 (0.84, 1.05)	1.02 (0.94, 1.11)	0.96 (0.90, 1.04)	1.14 (0.94, 1.37)	0.96 (0.64, 1.43)
0–5	0.99 (0.92, 1.06)	1.01 (0.92, 1.10)	0.95 (0.85, 1.07)	0.90 (0.75, 1.07)	1.02 (0.93, 1.10)	0.94 (0.77, 1.15)	0.95 (0.84, 1.07)	1.02 (0.93, 1.11)	0.96 (0.89, 1.04)	1.15 (0.94, 1.40)	1.09 (0.72, 1.66)
0–6	0.98 (0.91, 1.06)	1.00 (0.91, 1.11)	0.95 (0.84, 1.08)	0.90 (0.74, 1.09)	1.02 (0.93, 1.12)	0.90 (0.72, 1.13)	0.95 (0.84, 1.08)	1.01 (0.92, 1.12)	0.96 (0.88, 1.04)	1.15 (0.93, 1.43)	1.08 (0.66, 1.77)
0–7	0.99 (0.91, 1.07)	1.01 (0.90, 1.12)	0.96 (0.84, 1.09)	0.89 (0.73, 1.10)	1.03 (0.93, 1.13)	0.87 (0.69, 1.11)	0.95 (0.83, 1.09)	1.02 (0.91, 1.13)	0.96 (0.88, 1.05)	1.17 (0.93, 1.47)	0.97 (0.58, 1.61)
0–8	0.97 (0.89, 1.06)	0.99 (0.89, 1.11)	0.95 (0.82, 1.09)	0.89 (0.71, 1.11)	1.01 (0.91, 1.12)	0.89 (0.70, 1.14)	0.94 (0.81, 1.08)	1.00 (0.90, 1.12)	0.95 (0.86, 1.05)	1.15 (0.90, 1.47)	0.95 (0.55, 1.63)
0–9	0.98 (0.89, 1.07)	0.99 (0.88, 1.11)	0.96 (0.82, 1.11)	0.89 (0.70, 1.13)	1.00 (0.89, 1.12)	0.94 (0.73, 1.20)	0.94 (0.80, 1.10)	1.01 (0.89, 1.13)	0.95 (0.86, 1.05)	1.16 (0.90, 1.51)	0.93 (0.51, 1.68)
0–10	0.97 (0.88, 1.08)	0.99 (0.87, 1.12)	0.95 (0.81, 1.11)	0.91 (0.71, 1.16)	0.99 (0.88, 1.12)	0.95 (0.73, 1.23)	0.95 (0.81, 1.12)	0.99 (0.87, 1.13)	0.95 (0.85, 1.06)	1.19 (0.90, 1.56)	0.86 (0.45, 1.64)
0–11	0.96 (0.87, 1.07)	0.98 (0.85, 1.12)	0.94 (0.80, 1.11)	0.94 (0.73, 1.22)	0.97 (0.85, 1.10)	0.95 (0.72, 1.25)	0.96 (0.81, 1.14)	0.97 (0.85, 1.11)	0.94 (0.84, 1.05)	1.16 (0.87, 1.55)	0.89 (0.45, 1.77)
0–12	0.96 (0.86, 1.07)	0.97 (0.84, 1.12)	0.94 (0.79, 1.12)	0.98 (0.75, 1.29)	0.95 (0.83, 1.09)	0.96 (0.72, 1.29)	0.98 (0.81, 1.17)	0.95 (0.82, 1.10)	0.93 (0.83, 1.05)	1.15 (0.84, 1.58)	1.01 (0.49, 2.07)
0–13	0.95 (0.84, 1.07)	0.96 (0.83, 1.12)	0.93 (0.77, 1.12)	0.97 (0.73, 1.29)	0.93 (0.81, 1.07)	0.99 (0.73, 1.35)	0.97 (0.80, 1.17)	0.94 (0.81, 1.10)	0.93 (0.82, 1.05)	1.14 (0.81, 1.60)	1.02 (0.45, 2.29)
0–14	0.93 (0.82, 1.05)	0.94 (0.80, 1.11)	0.91 (0.75, 1.10)	0.90 (0.67, 1.22)	0.92 (0.79, 1.07)	0.99 (0.72, 1.36)	0.94 (0.77, 1.15)	0.93 (0.79, 1.09)	0.90 (0.79, 1.03)	1.13 (0.80, 1.61)	0.99 (0.42, 2.35)

All models adjusted for daily mean temperature and humidity. The odds ratio was calculated per 10-unit increase. ^a^ The warm season includes April–September. ^b^ The cold season includes October–March.

**Table 4 ijerph-18-00924-t004:** The association between short-term exposure to PM_2.5_ and children’s KD hospitalization in the two-pollutant models.

	Single Pollutant	Two-Pollutant Model			
Lag	PM_2.5_	Adjusted SO_2_	Adjusted NO_2_	Adjusted CO	Adjusted O_3_
0–1	1.01 (0.96, 1.06)	1.00 (0.94, 1.06)	1.03 (0.98, 1.10)	1.02 (0.95, 1.09)	1.01 (0.96, 1.07)
0–2	1.00 (0.95, 1.06)	0.99 (0.93, 1.06)	1.02 (0.96, 1.08)	1.02 (0.95, 1.09)	1.00 (0.94, 1.06)
0–3	0.99 (0.93, 1.05)	0.97 (0.93, 1.04)	1.00 (0.94, 1.07)	1.00 (0.93, 1.07)	0.98 (0.92, 1.05)
0–4	0.98 (0.92, 1.05)	0.97 (0.89, 1.04)	1.00 (0.93, 1.07)	0.98 (0.91, 1.06)	0.98 (0.91, 1.04)
0–5	0.99 (0.92, 1.06)	0.96 (0.89, 1.04)	0.99 (0.92, 1.07)	0.98 (0.91, 1.06)	0.98 (0.91, 1.05)
0–6	0.98 (0.91, 1.06)	0.97 (0.89, 1.05)	0.98 (0.91, 1.07)	0.98 (0.90, 1.07)	0.97 (0.90, 1.05)
0–7	0.99 (0.91, 1.07)	0.95 (0.87, 1.04)	0.98 (0.89, 1.07)	0.98 (0.90, 1.07)	0.98 (0.90, 1.06)
0–8	0.97 (0.89, 1.06)	0.94 (0.85, 1.03)	0.98 (0.89, 1.08)	0.97 (0.89, 1.07)	0.96 (0.88, 1.05)
0–9	0.98 (0.89, 1.07)	0.94 (0.85, 1.04)	0.97 (0.88, 1.08)	0.97 (0.89, 1.07)	0.96 (0.87, 1.06)
0–10	0.97 (0.88, 1.08)	0.95 (0.86, 1.05)	0.97 (0.87, 1.08)	0.96 (0.87, 1.07)	0.96 (0.87, 1.07)
0–11	0.96 (0.87, 1.07)	0.95 (0.85, 1.06)	0.97 (0.87, 1.08)	0.95 (0.86, 1.06)	0.95 (0.86, 1.06)
0–12	0.96 (0.86, 1.07)	0.93 (0.83, 1.05)	0.95 (0.85, 1.07)	0.94 (0.84, 1.06)	0.94 (0.84, 1.05)
0–13	0.95 (0.84, 1.07)	0.93 (0.82, 1.05)	0.94 (0.84, 1.06)	0.94 (0.83, 1.06)	0.94 (0.83, 1.05)
0–14	0.93 (0.82, 1.05)	0.91 (0.80, 1.04)	0.93 (0.82, 1.05)	0.93 (0.82, 1.05)	0.92 (0.81, 1.04)

All models were adjusted for daily mean temperature and humidity. The odds ratio was calculated per 10-unit increase.

## Data Availability

This data extracted Electronic Medical Record (EMR) system hospitalization data of Ewha Womans University Mokdong Hospital. The data are not publicly available due to ethical restrictions.

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
