# Peer review of "Is Short-Term Exposure to PM2.5 Relevant to Childhood Kawasaki Disease?"

_ijerph, 2021, doi:10.3390/ijerph18030924_

Round 1
Reviewer 1 Report
The authors presented a very elegant work investigating whether fine particulate matter (PM2.5) and other selected air pollutants contribute to Kawasaki Diseases (KD) hospitalization events based on a large group of KD children (N=771) in South Korea over 10 years (2006-2016).
The selection criteria for KD were well carefully elaborated and clearly presented in the paper.
The novelty of the paper is an attempt to investigate the role of PM 2.5 concentrations in the process of KD exacerbation (requiring hospitalization) development. Due to a lack of information on PM2.5 concentrations before 2015 PM2.5 data were calculated by applying ground and phase observations and modeling data. The modeling process was successfully used in the past and proofed to be sound.
The rich information about other than PM 2.5 air pollutants(PM10, SO2, NO2, CO, O3) was obtained from air pollution monitoring data. Exposure data were linked using administrative-district data based on the patients’ addresses, which is standard approached when individual dose estimation seems not to be possible.
The Authors investigate the acute relationship between short-term exposure to PM2.5 and KD hospitalization by bi-directional time stratification case-crossover design. The design is well described, which is of importance for readers with less experience in this method.
The results of the investigation of the association between KD and selected air pollutants (including rather high exposure to PM2.5 – 10 years average 30 ug/m3) proofed to be negative. The authors found no association between PM2.5 and children’s KD hospitalizations. However, three major air pollutants PM10, NO2, and SO2 had a positive but statistically insignificant association with KD incidence.
Minor remarks
1.I would suggest referring in the introduction or the discussion to a new syndrome resembling Kawasaki disease (vasculitis) in pediatric patients. The syndrome has been named a pediatric multisystem inflammatory syndrome (MIS-C). The syndrome was recently associated with SARS-CoV-2 infections in children. The information that KD is rather not associated with air pollution may be important in considering the etiology of MIS-C.
2) The term "hospitalization of KD" instead of "the incidence of KD" should be used as incidence refers to the development of new diseases.
Author Response
Thank you for the opportunity to revise our manuscript entitled “Is short-term exposure to PM2.5 relevant to childhood Kawasaki disease?” This response letter is our point-by-point responses to the comments raised by the learned reviewers. We would like to take this opportunity to express our sincere thanks to the expert reviewers who identified several areas of our manuscript that needed modifications. We also would like to thank you, for allowing us to resubmit a revised copy of the manuscript.
#Minor Comment 1
“I would suggest referring in the introduction or the discussion to a new syndrome resembling Kawasaki disease (vasculitis) in pediatric patients. The syndrome has been named a pediatric multisystem inflammatory syndrome (MIS-C). The syndrome was recently associated with SARS-CoV-2 infections in children. The information that KD is rather not associated with air pollution may be important in considering the etiology of MIS-C.”
Thank you for your comment. We agree with your suggestion. Therefore, we discussed the new syndrome resembling Kawasaki disease (vasculitis) in pediatric patients in the Discussion section. As you have pointed out, we feel the need to study the relevant content.
Before revision
None
After revision (In the Discussion section on page 2(actual page 12),in line 64–74)
Especially, since the beginning of the pandemic of coronavirus disease-2019 (COVID-19), a surge of SARS-Cov-2 patients with COVID-19 syndrome overlapping with KD, called multisystem inflammatory syndrome in children (MIS-C), has been reported in 2020.[1-3] KD and MIS-C share several common symptoms, such as skin rash, lymphadenopathy, strawberry tongue, coronary artery dilatation and an elevation inflammatory biomarkers such as C-reactive protein, procalcitonin, ferritin.[4] Furthermore, several unique symptoms of MIS-C and KD are prevalent in children, and children have fewer pro-inflammatory cytokines secretion and more active innate immune response than adults.[4] Therefore, understanding that the KD is rather not associated with PM2.5 may provide us with new information regarding the pathogenesis of MIS-C in COVID-19, as well as the conserve.
#Minor Comment 2
" The term "hospitalization of KD" instead of "the incidence of KD" should be used as incidence refers to the development of new diseases.”
Thank you for your comments. We agree with your suggestion. As you pointed out, we revised it as follows: “incidence of KD” -> “hospitalization of KD”.
Before revision
“This study analyzed the association between PM2.5 and the incidence of KD in children in South Korea over 10 years (2006-2016).”
“Our findings suggest that short-term exposure to PM2.5 was not significantly associated with an increased incidence of KD. We believe this is the first attempt to assess the impacts of PM2.5 on KD in South Korea.”
“The three major air pollutants PM10, NO2, and SO2 had a positive but statistically insignificant association with KD incidence. It was suggested that short-term exposure to high temperatures may significantly increase KD incidence, but the evidence linking air pollution and KD was limited”
After revision (In the Introduction section on page 2, in line 91–93,
In the Discussion section on page 2 (actual page 11–12))
“This study analyzed the association between PM2.5 and the hospitalization of KD in children in South Korea over 10 years (2006-2016).”
“Our findings suggest that short-term exposure to PM2.5 was not significantly associated with an increased hospitalization of KD. We believe this is the first attempt to assess the impacts of PM2.5 on KD in South Korea.”
“The three major air pollutants PM10, NO2, and SO2 had a positive but statistically insignificant association with KD hospitalizations. It was suggested that short-term exposure to high temperatures may significantly increase KD hospitalizaions, but the evidence linking air pollution and KD was limited”
Reviewer 2 Report
In the manuscript by Oh, et. al., the authors use statistical modeling approaches to examine potential relationships between PM2.5 exposure and incidence of Kawasaki disease. The effect of co-pollutants was also examined. This is a well done analysis which is of importance to the field.
One minor comment: The authors used a fusion of both observational and simulated data in their analysis. Were the end results the same when using only observational data?
Author Response
Title: Is short-term exposure to PM2.5 relevant to childhood Kawasaki disease?
Thank you for the opportunity to revise our manuscript entitled “Is short-term exposure to PM2.5 relevant to childhood Kawasaki disease?” This response letter is our point-by-point responses to the comments raised by the learned reviewers. We would like to take this opportunity to express our sincere thanks to the expert reviewers who identified several areas of our manuscript that needed modifications. We also would like to thank you, for allowing us to resubmit a revised copy of the manuscript.
#Minor Comment 1
“One minor comment: The authors used a fusion of both observational and simulated data in their analysis. Were the end results the same when using only observational data?”
Thank you for your comment. As described in the paper, there are restrictions that do not have an PM2.5 concentration before 2015. Therefore, we compared the modeling and observational data for the patients measured between 2015 and 2016.
The total number of patients during this study period is 162. The table below shows the estimated values per PM2.5 1-unit increment.
|
Lag |
CMAQ PM2.5 |
Monitoring PM2.5 |
||
|
Estimates |
Standard error |
Estimates |
Standard error |
|
|
0 |
-0.012 |
0.009 |
-0.013 |
0.010 |
|
0-1 |
-0.013 |
0.011 |
-0.017 |
0.012 |
|
0-2 |
-0.009 |
0.012 |
-0.011 |
0.013 |
|
0-3 |
-0.010 |
0.013 |
-0.012 |
0.014 |
|
0-4 |
-0.020 |
0.015 |
-0.021 |
0.016 |
|
0-5 |
-0.028 |
0.015 |
-0.028 |
0.017 |
|
0-6 |
-0.029 |
0.016 |
-0.031 |
0.018 |
Reviewer 3 Report
The study analyzes the association between short-term exposure to pm2.5 and the development of Kawasaki disease. To do this, the authors use a case-crossover design and data from a university hospital during the years 2006 to 2016. To assign the exposure, they use the Community Multiscale Air Quality model used in Korea. The justification for the study is based on the hypothesis that some environmental risks may be involved in the appearance of this disease. The results do not show a significant association between the variables studied even when adjusting for other variables of interest. The article is well structured and organized. I think it could be published after the authors answer the following questions, among which the one that concerns me the most is the exposure assignment .
My main doubts regarding the exposure assessment are:
- a) Was the exposure allocation disaggregated at the administrative district level according to the address or place of residence of each patient? I think this is not clear in the text.
- b) For the model that was used, was any validation exercise carried out with ground measurements ?
If that the case, what were the results of this validation?
- c) Regarding the measurements mentioned that was obtained from the monitoring network, could you explain a little more how you used them in the analysis? Was it used or not to assign the individual exposure of the people involved in the study?
- d) Do the authors consider that imprecision in the exposure asignment could lead to the non-association results? I think this point should be included in the discussion.
- e) Can you clarify in the next sentence, how much is too high?
“If the correlation coefficient 171 between the two exposure variables is too high”
- f) In table 2 could the results be included divided into case periods and control periods?
- g) Could you explain what was the method to assign the variables of temperature and humidity to each patient?
Other questions
In this paragraph
“In this design, the date settings for 147 the control group are the same year, month, and day of the weeks for the 148 KD hospitalization date, but the weeks differ.” You refer to the “control periods”, not the “control group”. If it is true, it must be corrected.
For the adjustment variables, I have the following questions:
- Why was it decided to categorize the ages in these groups? What is the justification?
- Stratification by season, is it not correlated with the variables of temperature and humidity? Was this situation controlled in the analysis?
One last question. For subsequent studies, the authors consider it appropriate to include the genetic conditions of the patients in the studies on air pollution and kawasaki disease?
Author Response
Title: Is short-term exposure to PM2.5 relevant to childhood Kawasaki disease?
Thank you for the opportunity to revise our manuscript entitled “Is short-term exposure to PM2.5 relevant to childhood Kawasaki disease?” This response letter is our point-by-point responses to the comments raised by the learned reviewers. We would like to take this opportunity to express our sincere thanks to the expert reviewers who identified several areas of our manuscript that needed modifications. We also would like to thank you, for allowing us to resubmit a revised copy of the manuscript.
#Major Comment 1-a)
"Was the exposure allocation disaggregated at the administrative district level according to the address or place of residence of each patient? I think this is not clear in the text.”
Thank you for your comment. We're sorry for any confusion and agree with your concern about our exposure assessment. Our exposure data is estimated by administrative districts, so we linked it to the administrative districts information of KD patients.
Before revision(None)
None
After revision (In the Method section on page 3, in line 135–138)
“The CMAQ PM2.5 data were linked using administrative-district data based on the patients’ place of residence. A total of 37 administrative districts are linked (Seoul: 17, Gyeonggi: 14, Incheon: 6).”
#Major Comment 1-b)
"For the model that was used, was any validation exercise carried out with ground measurements? ”
Thank you for your comment. The validation of the models used is previously described in the paper (titled: A Multiscale Tiered Approach to Quantify Contributions: A Case Study PM2.5 in South Korea During 2010-2017, doi:10.3390/atmos11020141). The study conducted a model evaluation using six super sites in Korea. This paper evaluates model performance for 9-km simulations, presenting R-square: 0.8, NMB: -19.2% normalized mean error (NME): 20.3%, and root mean squared error (RMSE)=6.2 μg/m3.
#Major Comment 1-c)
"Regarding the measurements mentioned that was obtained from the monitoring network, could you explain a little more how you used them in the analysis? Was it used or not to assign the individual exposure of the people involved in the study?”
Thank you for your comment. Multiple monitoring station are installed at each region. We selected and linked the nearest monitoring station from the KD patient's residence information.
Air pollution data has been assigned according to each patient's residence, so it can be seen as individual exposure. However, it is not a direct measurement but an exposure value adjacent to the monitoring station. We added the explain in the Methods section.
Before revision (In the Method section on page 4 in line 138–139)
“Exposure data were linked using administrative-district data based on the patients’ addresses.”
After revision (In the Method section on page 3 in line 143–145)
“Monitoring stations are installed in multiple locations by region. We selected and linked the nearest monitoring station from the KD patient's place of residence.”
#Major Comment 1-d)
"Do the authors consider that imprecision in the exposure assignment could lead to the non-association results? I think this point should be included in the discussion.”
Thank you for your comment. We agree with your suggestion. As you point out, inaccurate exposure assignment may not reveal the association between actual exposure (e.g. personal-level exposure) and health effects. Therefore, further studies need accurate individual level exposure measurements. We added the relevant sentence in the Discussion section.
Before revision(In the Discussion section on page 2 (actual page 13))
“First, we assigned municipality-level PM2.5 exposure data and did not measure or estimate individual-level exposure, including proximity to major roads or the height of each participant’s living place.”
After revision(In the Discussion section on page 3 (actual page 13), in line 90–94)
“First, we assigned administrative district-level PM2.5 exposure data and did not measure or estimate individual-level exposure, including proximity to major roads or the activities of each participant’s living place. Inaccurate exposure assignment may not properly reflect the relationship to health effects. Therefore, further studies need accurate individual level exposure measurements.”
#Major Comment 1-e)
“Can you clarify in the next sentence, how much is too high?”
If the correlation coefficient between the two exposure variables is too high
Thank you for your comment. We are sorry for any confusion. We presented a correlation coefficient in the supplementary files. As you pointed out, we modified the sentence.
Before revision(In the Method section on page 4 in line 171–173)
If the correlation coefficient between the two exposure variables is too high, it was excluded from the analysis.
After revision(In the Method section on page 4 in line 178–180)
If the correlation coefficient between the two exposure variables is too high (0.7 ≥), it was excluded from the analysis.
#Major Comment 1-f)
" In table 2 could the results be included divided into case periods and control periods?”
Before revision
|
Mean |
SD |
Median |
IQR |
|
|
PM2.5 (μg/m3) |
33.5 |
20.9 |
29.2 |
21.4 |
|
Mean temperature (°C) |
12.7 |
11.1 |
13.8 |
20.1 |
|
Mean humidity (%) |
62.4 |
14.9 |
62.8 |
21.4 |
After revision
|
Exposure variables |
Case periods (N= 771) |
Control periods (N= 2,629) |
Mean difference |
95% confidence interval |
||
|
Mean |
SD |
Mean |
SD |
|||
|
PM2.5 (μg/m3) |
34.13 |
22.37 |
33.33 |
20.46 |
0.80 |
-0.96, 2.57 |
|
Mean temperature (°C) |
12.82 |
10.93 |
12.70 |
11.19 |
0.12 |
-0.76, 1.00 |
|
Mean humidity (%) |
62.57 |
14.83 |
62.34 |
14.86 |
0.23 |
-0.97, 1.41 |
#Major Comment 1-g)
"Could you explain what was the method to assign the variables of temperature and humidity to each patient?”
Thank you for your comment. We are sorry for any confusion. We got the temperature/humidity data released by the Korea Meteorological Administration. Since the temperature and humidity data are open by city, we linked the data to patients by region (Seoul, Busan, Incheon).
Before revision(In the Method section on page 3 in line 142–143)
The mean average temperature and humidity were linked to each patient’s residence in each region.
After revision(In the Method section on page 4 in line 148–150)
Since the temperature and humidity data are open by city, the daily mean temperature and humidity were linked to each patient’s residence cities (Seoul, Busan, Incheon).
# Minor comment 1
" “In this design, the date settings for the control group are the same year, month, and day of the weeks for the KD hospitalization date, but the weeks differ.” You refer to the “control periods”, not the “control group”. If it is true, it must be corrected.”
Thank you for your comment. We have revised the sentences to avoid confusion.
Before revision(In the Method section on page 3 in line 147–152)
In this design, the date settings for the control group are the same year, month, and day of the weeks for the KD hospitalization date, but the weeks differ. Therefore, three or four controls can be assigned to one KD patient. In this study design, because each case or control group is the same patient, personal characteristics such as sex and age are not considered in the short-term fluctuations.
After revision(In the Method section on page 4 in line 154–158)
In this design, the date settings for the control periods are the same year, month, and day of the weeks for the KD hospitalization date, but the weeks differ. Therefore, three or four controls can be assigned to one KD patient. In this study design, because each case or control periods the same patient, personal characteristics such as sex and age are not considered in the short-term fluctuations.
# Minor comment 2
"Why was it decided to categorize the ages in these groups? What is the justification?”
Thank you for your comment. Kawasaki disease is an acute febrile vascular disease that occurs in children. Kawasaki disease occurs in infants and young children, mainly under the age of five. Our study patients included 10 years of age or younger. Since little is known about the association between Kawasaki disease and PM2.5, we wanted to know the difference by age. Another reason is that the level of biological development varies depending on age.
# Minor comment 3
"Stratification by season, is it not correlated with the variables of temperature and humidity? Was this situation controlled in the analysis?”
Thank you for your comment. Temperature and humidity were adjusted in the main model as confounding factors, and similarly in seasonal stratification analysis.
# Minor comment 4
"One last question. For subsequent studies, the authors consider it appropriate to include the genetic conditions of the patients in the studies on air pollution and kawasaki disease?”
Thank you for your comment. We agree with your suggestion. We added the relevant sentence in the Discussion section.
Before revision
None
After revision (In the Discussion section on page 3 (actual page 13), in line 101–105)
Finally, the further studies including genetic susceptibility between air pollution and KD would be worthwhile to evaluate the immune environment and its response in the body. Because KD occurs most often in children from northeast Asia. Even in the United States, KD occurs most often among children of Asian-American or Pacific Island ancestry.[5]
Round 2
Reviewer 3 Report
From my point of view, the authors have responded adequately to my observations. I have no additional comments